

**Effects of long-term mowing on the fractions and chemical composition of soil organic matter in a semiarid grassland**

Jiang-Ye Li[1], Qi-Chun Zhang[1], Yong Li[1] and Hong-Jie Di[1]

[1]Zhejiang Provincial Key Laboratory of Agricultural Resources and Environment, Key Laboratory of Environment Remediation and Ecological

Health, Ministry of Education, Zhejiang University, Hangzhou 310058, China

*Correspondence to:* Qi-Chun Zhang (qczhang@zju.edu.cn)

**Abstract.** Grassland is the second largest carbon pool following forest. Intensive mowing is common to meet the need of increased livestock. However, little information on the quality and quantity of soil organic matter (SOM) under different mowing managements was documented. In this work, the fractions and chemical composition of SOM under different mowing managements were studied using traditional fractionation

method and spectroscopy technology ([13]C-NMR and FTIR) based on a 13-year mowing trial with four treatments: control (CK, unmown), mowing once every second year (M1/2), mowing once a year (M1) and mowing twice a year (M2). The results showed that M1/2 and M1 significantly enhanced the SOM accumulation while M2 did not significantly impacted SOM content but it significantly limited the SOM humification and degradation. Substituted alkyl carbon (C) was the major organic C type as it made up over 40% of the total C. M1/2 and M1 significantly increased stable C functional groups (alkyl C and aromatic C) by degrading labile C functional group (*O*-alkyl C) and forming calcium humic acid while M2

had opposite effects. The increase of NMR indices (HB/HI, Al/Ar, A/OA and CC/MC) in M1/2 and M1 further suggested that M1/2 and M1 increased the stability of SOM. Significant correlations between net N mineralization or MBC and C functional groups indicated that the shifts of SOM fractions and chemical composition were closely related to soil microbial activity. Meanwhile, M1 significantly increased soil MBC while M2 worked oppositely. Therefore, M1 are the most recommended mowing management while M2 should be avoided in the semiarid grassland.

**Key words:** Mowing, Soil organic matter composition, Solid state CPMAS [13]C-NMR, grassland



## 1 Introduction

Soil organic matter (SOM) plays a central role in the global biogeochemical cycles of most major nutrients and soil organic carbon (SOC) is a major factor for $CO_2$ concentrations in the atmosphere. Grassland accounts for more than 40% of the terrestrial area globally and in China (White et al., 2000; China's Environmental Bulletin, 2006). It has been estimated that 89% of C sequestrated by photosynthesis in grassland ecosystems is stored in the soil, and C storage in grassland soils accounts for 26% of the total global terrestrial C storage (DeFries et al. 1999;

Schlesinger et al., 1997). The Inner Mongolia grasslands cover over 70% of the total land area of the region, and represent more than a quarter of the total grassland area in China. Mowing in the autumn, a widely used practice of grassland for preparing winter feed for livestock, is reported to facilitate plant species richness (Socher et al., 2012) and further increases the soil carbon (C) and nitrogen (N) stocks caused by enhanced plant productivity, root biomass and root exudates (Cong et al., 2014). However, high frequency mowing to meet the needs of increased numbers of livestock may result in opposite effects on plant species diversity and block soil C and N turnover as few microbes were able to bear such a

degree of disturbance.

Increased plant diversity and enhanced fresh SOC input can lead to the degradation of recalcitrant organic compounds by priming effect (Fontaine et al., 2011). In addition, different plant species release diverse organic compounds and these would impact on soil microbial communities (Dijkstra et al., 2005). Mowing once a year can increase the activity of extracellular enzymes to decompose polymeric C (aromatic polymer existing in lignin and cellulose derived from litter or root residue) into monomers composed by simple but resistant C like alkyl C

(Steinauer et al., 2015). However, it is unclear how stable the SOM is under the different mowing managements. The stability of soil C pool is closely related to the sustainability of soil functions. Therefore, to better assess the ecological significance of long-term mowing managements, it is necessary to study the impacts of different mowing managements on the quantity and quality of soil organic matter.

Soil organic matter (SOM) composition is often used to evaluate the stability of soil C pool. The fractions of SOM are classified according to their particle size and bioavailability by traditional methods (Six et al., 2002). Spectroscopy is a powerful tool for identifying the chemical

structure of SOM as soil samples are measured directly rather than determined after a series of extractions which might alter the nature of SOM. Fourier-transform-infrared-spectroscopy (FTIR) and nuclear magnetic resonance (NMR) are widely used to study the chemical composition of organic matter (Olk et al., 2008; Mao et al., 2012; Zhou et al., 2014). The application of advanced solid-state NMR, i.e. cross-polarization magic angle spinning [13]C-NMR (CPMAS [13]C-NMR) in characterizing chemical structures of SOM, is also an important approach to reveal the essential changes of SOM formation and degradation in different ecosystems (Zhou et al., 2014; Panettieri et al., 2014; Zhang et al., 2015) and in litter

decomposition processes (Sanaullah et al., 2012; Bonanomi et al., 2013). This approach can provide the information of SOM structure noninvasively without using solvents. However, the cost of solid-state NMR is high, especially for complex soil samples, because of the length of time it takes to identify the chemical structure.

To better understand and evaluate the nature of SOM, elemental analysis (EA) and FTIR are often combined to help get more accurate information (Mao et al., 2008; Zhou et al., 2015). In grassland ecosystems, these tools are also used to study SOM stocks and quality (Baumann

et al., 2013; Knicker et al., 2012). However, there are no reports of the effects of mowing and mowing frequency on the chemical structure of the whole SOM. In this study, we combined advanced solid-state NMR with traditional methods to investigate the quality and quantity of the grassland soil organic C under different mowing managements. The objective of this study was to investigate the impacts of long-term mowing practices on the chemical composition of SOM and evaluate the stability of the grassland soil carbon pools under different mowing frequencies.

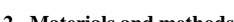

## 2 Materials and methods

### 2.1 Site description and experimental design

The study site is located in the Xilingol region of Inner Mongolia (43 ′269′N – 44°089′N and 116°049′E – 117°059′E) in northern China. It has a temperate semiarid climate, with an annual mean temperature of 0.5 ℃ and annual average precipitation of 350 mm, most of which falls during the summer. The annual potential evaportranspiration ranges from 1,600 to 1,800 mm. The soil is Calcic-orthic Aridisol according to the US soil taxonomy (or sandy-loam dark chestnut soil in the Chinese classification system) (Baoyin et al., 2014) and in the profile, there is a humus layer of 20 – 30 cm and a calcic layer at ca. 50 cm depth (Jiang et al., 1988). The characteristic vegetation of this region is *Leymus chinensis* (*L. chinensis*), accounting for $55 \pm 15$ % (mean ± standard deviation) of total herbage yield. Other species in the order of decreasing proportion of total herbage yield are tall bunchgrasses (mostly *Stipa grandis* and *Agropyron michnoi*), short bunchgrasses [*Cleistogenes squarrosa* and *Koeleria cristata* (L.) Schrad] and sedge (*Carex korshingski* Kom.), forbs and legumes (Baoyin et al., 2014). The growing season usually starts in May and ends in September.

The long-term mowing experiment began in 2001 in a permanent enclosure by the Inner Mongolia Grassland Ecosystem Research Station of the Chinese Academy of Sciences. The enclosure for the mowing experiment was divided into 15 plots (24 m × 20 m for each plot). There were five treatments with 3 mowing treatments and the Control (CK, unmown), each with 3 replicates. The 3 mowing treatments were described in Table 1. The grasses were cut at a height of 6 cm and all the cuttings were removed from the plots. A 2 – 4 m buffer zone was designed between plots. The mowing dates might change slightly in different years based on the growth rates of *L. chinensis*, and the mowing dates of 2[nd] Jun., 16[th] Aug. and 12[th] Sep. represent the time when the grass palatability is best for the animal, the aboveground biomass reaches the peak and the grass stop growing, respectively.

### 2.2 Soil samples collection

Soil samples were collected from 0 – 10 cm depth using a soil auger (7 cm in diameter and 10 cm in depth) in October, 2013 at the end of the growing season. Five soil cores were collected from each plot at random locations and they were combined and mixed thoroughly in each plot. Visible roots and litter residues and large soil fauna in the soil samples were removed. The soil samples of around 1 kg were put into ziplock bags and transformed to the lab on ice quickly. In the lab, the soil samples were passed through a 2-mm sieve and homogeneously mixed again and then kept at 4 ℃ soil sample was before the chemical analysis.

### 2.3 Measurements of bulk soil basic properties

Air-dried soils were analyzed for pH using a water to soil ratio of 2.5:1. Soil moisture content was determined by oven drying to a constant mass at 105 ℃ for 16 h. The content of soil organic C (SOC) and total nitrogen (TN), alkali-hydrolyzable N (AN), Olsen phosphorus (Olsen P) and net N mineralization were determined, referring to Kalembasa and Jenkinson (1973), Bremner (1965), Bao (2000) and Lin (2010), respectively.

### 2.4 Soil organic matter fractionation

Soil microbial biomass carbon (MBC) was determined using the chloroform fumigation extraction method (Vance et al., 1987; Wu et al., 1990). Water soluble organic carbon (WSOC) was determined using a modified method (Li et al., 2013). Briefly, they were extracted from 5.0 g of fresh soil using a soil to water ratio of 1:10 at 25 ℃, and shaken for 30 min at a speed of 250 rpm. The samples were subsequently centrifuged for 20 min at 2500 rpm, and then the supernatant was filtered using a 0.45 μm membrane filter. The filtrate was measured by the same TOC analyzer. Soil readily oxidizable carbon (ROC) was determined and calculated following the detailed procedure described in Li et al., (2013). The mobile humic acid (MHA) and calcium humic acid (CaHA) were extracted following the procedure by Mao et al. (2008). Thirty g of air-dry



soil were used to extract the two humic fractions. Finally, the fractions were dried by freezing-drying (FD-1C-50, Beijing, China).

**2.5 Analysis of the chemical composition of soil organic matter (SOM)**

To remove paramagnetic materials ($Fe^{3+}$, $Mn^{2+}$) and increase the signal-to-noise ratio, the soil samples were pretreated with HF (10%, $v/v$) using the procedure detailed in Li et al., (2010), and finally, the SOM samples were freeze-dried. It is reported that the chemical composition of SOM was not altered as the C/N was similar before and after the HF processing (Mao et al., 2008; Zhou et al., 2014), as was the case in this study (Table 2 and S2).

**2.5.1 Elemental analysis**

The elemental composition of SOM samples was determined by dry combustion, followed by gas chromatography using a CHNS Elemental Analyzer (Carlo Erba model EA1108, Italy Vario). The content of O was estimated as the ash-free mass less C, H, and N. Ash content was determined by combustion overnight in a muffle furnace at 500 °C (Ussiri and Johnson, 2003).

**2.5.2 FTIR analysis**

The FTIR analysis of the SOM samples was conducted on an Avatar 370 FTIR spectrometer (Thermo Nicolet, America). Each sample was prepared by grinding 2 mg freezing-drying SOM sample with 200 mg oven-dried KBr in a vibrating puck mill and then about 150 mg mixtures were compressed into a translucent pellet using a hydraulic compressor. The pellet was immediately placed on the sample holder, and all spectra ranging from 4,000 to 400 $cm^{-1}$ were recorded under the conditions of 4 $cm^{-1}$ wave number resolution, 25 scans, and pure KBr spectra as background (Zhou et al., 2014). Absorption peaks or bands were assigned to organic functional groups following Zhou et al., (2014). Only peaks

or bands in the functional group region from 4,000 to 1,000 $cm^{-1}$ of FTIR spectra were assigned because peaks in the fingerprint region below 1,000 $cm^{-1}$ were difficult to assign and were very complex, usually overlapping with signals of inorganic soil minerals.

**2.5.3 Solid state CPMAS $^{13}$C-NMR analysis**

Solid NMR experiment was performed on a Bruker Avance II 300 (Bruker Instrumental Inc) equipped with a 7 mm CPMAS (cross-polaration magic-angle-spinning) detector. NMR spectra were acquired under the conditions of a spectrometer frequency of 75 MHz, a MAS spinning

frequency of 5,000 Hz, a recycle time of 2.5 s and a contact time of 2 ms. The external standard used for chemical shift determination was hexam-ethylbenzene (methyl at 17.33 ppm). The quantified contribution of each type of C to the total signal intensity and promotion in CPMAS $^{13}$C-NMR spectrum was automatically integrated to calculate the area of the peaks which appeared in the corresponding chemical region using MestreNova software 8.1.0 (Mestrelab, Research Inc).

**2.6 Statistical analysis**

Data was statistically analyzed using SPSS 21.0 by one-way analysis of variance (ANOVA), and means were separated by Duncan's multiple range test at 5% level. The values (except for the ratios) reported in the tables are means ± standard error (S.E.). The figures were created using Origin 8.1. Linear regression analysis was conducted after the Pearson product-moment correlation analysis by two-tailed test in SPSS 21.0.

**3    Results**

**3.1 Basic properties of bulk soil and net N mineralization**

Soil pH was around 7.3 and was little affected by long-term mowing (Table 3). However, long-term mowing had a significant impact on soil nutrient concentrations. Compared with CK (unmown), mowing once every second year (M1/2) and mowing once a year (M1) significantly



increased SOC content ($P$ <0.05) while the SOC content in M2 was similar to that in CK ($P$ >0.05). The TN content in treatment M1 was the highest and significantly higher than that in treatment M2. The total N content in M2 was also significantly lower than those in the other two treatments (M1/2 and CK). Soil Olsen P contents in all the treatments were very low, around 1.2 mg kg$^{-1}$, and no significant difference was

observed between the treatments ($P$ >0.05). The AN content in the soil in M2 was significantly lower than those in the other treatments ($P$ <0.05) while there was no significant difference between the other treatments ($P$ >0.05). Net N mineralization in M1 was significantly greater than that in the other treatments, and it was significantly lower in M2 than that in other treatments ($P$ <0.05).

**3.2 Soil organic matter fractions**

Long-term mowing had a major impact on MBC (Table 4). The content of MBC in all treatments showed that M1/2 > M1 ≈ CK > M2.

Compared with CK, mowing significantly decreased soil WSOC content by more than 50% ($P$ <0.05). However, no difference was found in soil WSOC among the mowing treatments. The average content of ROC in the soils in mowing treatments was less than 50% of that in CK. In the mowing treatments, ROC content in M1 was significantly higher than that in M2 ($P$ <0.05)

The total content of both humic fractions (MHA and CaHA) accounted for a major proportion of SOM, especially in M1 where it reached 73.0% (Table 4), and this was significantly higher than that (53.1%) in M2 ($P$ <0.05). The content of CaHA was about 2 – 4 times that of MHA

across all treatments. MHA was recognized to be more labile than CaHA for microbes in soil (Mao et al., 2008). Compared to CK, M2 significantly decreased MHA content ($P$ <0.05), but did not affect CaHA content significantly ($P$ >0.05). However, M1 and M1/2 significantly increased CaHA content ($P$ <0.05) but did not significantly affect MHA content. Thus, both MHA and CaHA contents in soils of M1/2 and M1 were significantly higher than that in M2 ($P$ <0.05).

**3.3 Chemical composition of SOM**

Parameters of the elemental composition of the SOM were shown in Table 5. The content of hydrogen (H) and oxygen (O) varied from 0.49 – 0.63% and 0.25 – 0.35%, respectively. Compared with CK, mowing significantly decreased the H content and mowing twice a year (M2) also significantly decreased O content ($P$ <0.05). The ratio of H/C and O/C varied from 0.13 – 0.16% and 0.06 – 0.09%, respectively, and the H/C and O/C ratio in M1/2 and M1 were significantly decreased compared with M2 or CK.

The FTIR spectra of the SOM extracted from the grassland soil under different mowing treatments was shown in Fig. 1. The spectra were

dominated by the broad peak around 3,406 cm$^{-1}$, sharp peaks around 1,030 cm$^{-1}$ and medium sharp peaks around 1,653 cm$^{-1}$, which were ascribed to O-H stretching in alcohols, carboxylic acids and phenols, C-OH stretching in carbohydrates, and C=C stretching in aromatics, respectively. The intensity of other peaks in the FTIR spectra was relatively low. Small peaks at 2,928 and closing to 1,500 cm$^{-1}$ due to aliphatic C-H stretching in CH$_2$/CH$_3$, amide N-C/amino-NH vibrations, and aliphatic C-H bending in CH$_2$/CH$_3$, respectively, were found in all treatments. However, the FTIR spectra did not show clear differences of the chemical structure of SOC between CK and the mowing treatments or between

all the mowing treatments. The obvious differences among treatments appeared at the dominant peaks at 1,030 cm$^{-1}$, which showed that higher intensity with increasing mowing frequency, indicating that mowing significantly decreased the concentrations of carbohydrates. The peak intensity at 1,030 cm$^{-1}$ in M2 was the strongest, which might be because C-OH stretching in carbohydrates was decomposed slowly by fewer microbes (Fig.1 and Table 4).

The spectra of $^{13}$C-NMR is significantly grouped into four regions representing four main chemical environments of a $^{13}$C nucleus: carbonyl

C (210 – 160 ppm), aromatics C (160 – 110 ppm), substituted alkyl C (110 – 45 ppm), alkyl C (45 – 0 ppm) (Plaza et al., 2013; Zhao et al., 2012; Boeni et al., 2014). The carboxyl and carbonyl C (210 – 160 ppm), were allocated in aldehyde C, ketonic C and quinine C (210 – 185 ppm) and carboxyl C (175 – 165 ppm); aromatics C (165 – 110 ppm), was separated into phenolic C (160 – 140 ppm) and aryl-C (140 – 110 ppm); substituted C (110 – 45 ppm) was further divided into di-$O$-alkyl C (110 – 90 ppm), $O$-alkyl C (90 – 60 ppm) and $N$-akyl/methoxyl C (60 – 45

ppm) (Baumann et al., 2009, 2013; Zhao et al., 2012). Di-*O*-alkyl C includes compounds of polysaccharides of anomeric C (C1). *O*-alkyl C

represents carbohydrates, including compounds of C2 – C6 carbohydrates. *N*-akyl/methoxyl C includes lignin residues and peptide residues

amino acids and proteins. The labels used to refer to each chemical region are thought to be representative of the major forms of carbon present

in that region. Fig. 2 showed the $^{13}$C-NMR spectra of the SOM extracted from the grassland soil with different mowing managements (Fig. 2A)

and the detailed C functional groups represented by the peaks in the $^{13}$C-NMR spectra (Fig. 2B) were shown. One exception to this is the *N*-alkyl

region in which methoxyl carbon (present e.g. in lignin) may make a significant contribution to the measured signal intensity; however, for this

paper the 60 – 45 ppm region will be labelled as *N*-alkyl. Main C-types found in important biomolecules are listed in Table S1 Previous studies

have indicated that variations in the relative areas of signal peaks within a given spectral region which are >2.0% can be considered significant

for NMR quantitative analyses at the level of $P$ <0.05 (Baldock and Smernik, 2002) and it has been shown that there were no significant

differences in SOC and other major chemical properties for the three replicates of each treatment. We only used a single soil sample for each

treatment to prepare SOM samples for NMR and FTIR analysis, consistent with Zhou et al., (2014).

In all spectra, the alkyl C (45 – 0 ppm) and substituted alkyl C (110 – 45 ppm) peaks were dominant components in SOC composition across

all the treatments, accounting for 24.6 – 27.9% and 41.5 – 47.6% of the total spectral fractions, respectively (Table 6 and Fig. 2), followed by

aromatic C (160 – 110 ppm) and carbonyl C (210 – 160 ppm) peaks, accounting for 16.3 – 19.1% and 9.3 – 13.7% of the total spectral fractions,

respectively. In the substituted alkyl C, *O*-alkyl C (90 – 60 ppm) was the main fraction, making up more than 50% of the substituted alkyl C

while di-*O*-alkyl (110 – 90ppm) only accounted for less than 21% of the substituted alkyl C, and *N*-alkyl or methoxyl C were medium.

Compared to CK (unmown), mowing significantly increased alkyl C (except for M2) and significantly decreased substituted alkyl C ($P$ <0.05),

mainly existing in carbohydrates. The *O*-alkyl C in M2 was the lowest among all treatments, which was also consistent with the results of FTIR.

However, in the substituted alkyl C, mowing significantly increased *N*-alkyl or methoxyl C ($P$ <0.05). The proportion of aromatic C (aryl and

*O*-aryl C, 160 – 110 ppm) in M1/2 and M1 was significantly higher than that in CK while the proportion of carbonyl C (210 – 160 ppm) in these

two treatments was significantly lower than that in CK ($P$ <0.05). Mowing twice a year significantly increased the percentage of *O*-aryl C and

carbonyl C (210 – 160 ppm, $P$ <0.05).

     To better evaluate the quality of C pool, some indices were calculated following the formula in Table 2 and the results were shown in Table 7.

Compared with CK, mowing treatments significantly increased soil lignin C, A/OA ratio and HB/HI ratio while it significantly decreased L/P

ratio and CC/MC ratio ($P$ <0.05). There was little or no difference in aliphaticity and aromaticity among all treatments ($P$ >0.05), but the Al/Ar

ratio in M2 was significantly lower than that in the other treatments ($P$ <0.05). Compared with CK and M2, soil substituted C in M1/2 or M1 was

significantly higher ($P$ <0.05). On the contrary, the L/P ratio of SOC in treatment M1/2 and M1 was significantly lower than that in CK, but this

index in treatment M2 was significantly higher than that in CK ($P$ <0.05).

### 3.4 Correlation between net N mineralization and SOM fractions or the C functional groups of SOM

Soil organic matter content was significantly and positively correlated with MBC, MHA, CaHA and net N mineralization with r = 0.45, 0.48,

0.89, 0.54 ($P$ <0.05), but not correlated with WSOC and ROC ($P$ >0.05) (Table 8). ROC was significantly correlated with WSOC, MBC, MHA,

CaHA and net N mineralization (r = 0.55 – 0.90, $P$ <0.05). Moreover, positive correlations were found between net N mineralization and MBC,

ROC, MHA with correlation coefficients of 0.60, 0.91, 0.83, respectively, while negative correlations were found between net N mineralization

and CaHA (r = − 0.75, $P$ <0.05).

     The relationships between net N mineralization or MBC and the C functional groups of SOC are shown in Table 9. The results showed that N

mineralization was related to the chemical structure of SOC and to microbial biomass. Net N mineralization was not significantly related to five

detailed CPMAS $^{13}$C-NMR regions (*N*-alkyl/methoxyl C, *O*-alkyl C, di-*O*-alkyl C and aryl C), with r = 0.28, 0.37, 0.47 and − 0.24, respectively

($P$ >0.05), but was negatively correlated to *O*-aryl C (r = − 0.94, $P$ <0.001) and carbonyl C (r = − 0.79, $P$ <0.01) and the integrated aromatics



including aryl C and *O*-aryl C (r = −0.81, *P* <0.01). Consistent with net N mineralization, significant negative correlations were also found between MBC and *O*-aryl C (r = − 0.84, *P* <0.001), carbonyl C (r = − 0.96, *P* <0.01) and the integrated aromatics (r = − 0.39, *P* <0.05). However, both net N mineralization and MBC were positively correlated to alkyl C with r = 0.46 and 0.59, respectively (*P* <0.05). Different from net N mineralization, MBC was also significantly correlated with di-*O*-alkyl C and aryl C with r = 0.59 and 0.73 (*P* <0.05), respectively, but not correlated with *N*-alkyl/methoxyl C and *O*-alkyl C.

## 4 Discussion

### 4.1 Shifts of SOM content under different mowing managements

The overall and significant increased trend of SOM in the surface soil (0 – 10 cm) was shown in the moderate mowing frequency treatments (M1/2 and M1), while a significant decreased SOM was shown in heavy mowing treatment in this study (Table 3 and 4). Some studies showed that mowing enhanced plant species by increasing the subordinate plants (Marriotte et al. 2015; Socher et al., 2012) which may, in turn, influence soil N cycle. On the one hand, enhanced plant species richness promoted plant productivity and thus increased soil carbon and nitrogen stocks in grasslands without legume by more input of organic C and N from root biomass, root exudates and N retention (Cong et al., 2014). On the other hand, legume was common in grassland and its productivity increased due to mowing, thus increasing N fixation (Cardinale et al., 2012). Enhanced C and N stocks had a positive feedback to plant productivity, because soil N mineralization was increased (Chong et al., 2014) (Table 4). This study showed that mowing once a year significantly enhanced soil net N mineralization (Table 8), but excessive mowing practice resulted in herbage productivity decline due to high nutrient removal from the soil (Baoyin et al., 2014). Therefore, microbial biomass decreased by 28% in M2 compared to CK and soil net N mineralization was reduced significantly (Table 4).

The changes of SOC content in different treatments were mainly due to the changes of MBC, MHA and CaHA (Table 8). Microbes are sensitive to perturbation and thus MBC is regarded as a reliable indicator of the change of SOC pools by management practices (Fang et al., 2009). MHA and CaHA are two humic fractions, both rich in nutrients. However, MHA is younger and more labile than CaHA (Olk, 2006; Mao et al., 2008) and thus these two fractions behave differently in N cycling (Table 8). It is interesting that ROC, a labile C pool, was significantly related to CaHA. This might be because ROC from plants enhanced the microbial activities in the process of forming humus. The significant and positive linear correlation between net N mineralization and MBC in this study might suggest that enhanced microbial activity produced more extracellular enzymes to degrade organic N. On the whole, moderate mowing not only promoted the degradation of active SOM but also enhanced humification of SOM, and led the accumulation of SOM due to the increase of plant productivity (Baoyin et al., 2014) and microbial biomass (Table 4).

### 4.2 Stability of SOM impacted by different mowing treatments

Different mowing treatments had different impacts on the chemical structure of SOM. The stability of SOM reflects the difficulty of SOM degradation. The elemental analysis showed that mowing had major impacts on the chemical composition of SOM. The lower atom H/C ratio indicates more aromatic compounds or higher aromaticity and saturability, and the higher atom O/C ratio indicates more carboxyl groups, phenol or carbohydrates with oxygen (Ma et al., 2001; Steelink et al., 1985; Kim et al., 1991). Thus, the results of elemental analysis showed that long-term moderate mowing managements (M1/2 and M1) significantly increased the aromaticity but significantly decreased carboxyl and phenol groups or carbohydrates with oxygen. These indicated that moderate mowing practices increased the stability of SOM as aromatic compounds and phenols are stable constituents in the soil. The quantified analysis of ¹³C NMR spectra also directly showed that aromatics (160 – 110 ppm) increased in the moderate mowing treatments (Table 6). This is probably ascribed to the increased plant and microbial residues because moderate mowing could increase plant diversity and productivity (Marriotte et al. 2015; Chong et al 2014). Aryl C at 140 – 110 ppm is rich in condensed aromatics including charcoal, which is quite stable in soil (Zhou et al., 2014). The concentration of aromatics (160 – 110 ppm)



was significantly higher in all mowing treatments than that in CK, indicating that the humification process was enhanced by mowing. Moreover,

moderate mowing managements improved the CaHA fraction by 46.9 – 52.5% after 12 years. These all suggested that long-term moderate mowing managements enhanced the degree of humification of SOM as lignin, an important precursor substance for humus formation, increased in moderate mowing treatments (Table 7). Quite different from the moderate mowing managements, both the results of elemental analysis and $^{13}$C-NMR spectra showed that higher frequency mowing (M2) had little impact on the aromaticity of SOM (Table 6 and 7) and even limited the degree of humification (Table 4). In the M2 treatment, microbial biomass was significantly reduced (Table 3). These suggested that M2 had a

negative effect on microbial activity and soil C turnover, because the plant diversity and biomass were limited (Socher et al., 2012; Mariotte et al., 2013), resulting in lower labile carbon (Table 4).

In the CPMAS $^{13}$C-NMR indices (Table 7), A/OA (alkyl C/$O$-alkyl C) ratio is generally taken as a sensitive index of the extent of decomposition as alkyl C is a biodegraded product of the $O$-alkyl C (Baldock et al., 1997). $O$-alkyl C is easily decomposed while alkyl C is recalcitrant. In general, alkyl C and $O$-alkyl C keep a trade-off relationship (Li et al., 2013). The A/OA ratio in mowing treatments was

significantly higher than that in CK. When the value of A/OA ratio is relatively high, it indicates that the degree of decomposition of SOM is high. It can be considered that SOM is difficult to be further decomposed (Zhao et al., 2012). Therefore, moderate mowing (M1/2 and M1) enhanced the stability of SOC pools as well as primed the degradation of liable C pools. This might happen as increased microbes synthesized some recalcitrant components while decomposing the easily breakable components. The carbonhydrate C/methoxyl C (CC/MC) ratio is a new indicator to reflect the degree of degradation of SOM, (Mather et al., 2007), and both of CC/MC and A/OA ratios could indicate the degradation

of SOM in our study. The aliphaticity/aromaticity (Al/Ar) ratio is a predictor to reveal the complexity of the chemical composition of SOC, and the higher the value, the simpler the chemical composition of SOC. The hydrophobic C/hydrophilic C (HB/HI) ratio was used as a measure of chemical recalcitrance of C in the SOM extracted by HF, and the higher this value, the more stable the SOC (Boeni et al., 2014). The increased HB/HI ratio manifested that SOC in moderate moving treatments is more recalcitrant to be mineralized (Boeni et al., 2014) while Al/Ar ratio did not well reflect the complexity of the chemical composition of SOC in moderate mowing, but it showed that heavy mowing decreased the

complexity. From the viewpoint of C functional groups, moderate mowing managements improved the stability of SOC, which is closely associated with the increase in recalcitrant compounds rich in alkyl C (waxes, resin, cutin, suberin, peptide side-chain, long-chain aliphatics), mainly derived from the increased plant materials (roots and litter) (Socher et al., 2012; Mariotte et al., 2013), accompanied by preferential degradation of $O$-alkyl C of carbohydrates, polysaccharides and other liable C (Table S1), and increases in aromatic C (lignin), and cellular residues of microbes (Table 4). On the contrary, these indices in the higher frequency mowing treatment (M2) were similar to those in CK. It is

interesting that lignin C in treatment M2 was significantly higher than that in moderate mowing treatments and CK (Table 7), which might be because heavy mowing limited the growth of microbes degrading lignin. The methoxyl C is also recalcitrant C and it is relatively enriched in topsoil when $O$-alkyl or di-$O$-alkyl C is prone to oxidation. The methoxyl C is only from lignin (Zhao et al., 2012). Mowing once a year (M1) also greatly increased the methoxyl C and then improved the stability of SOM.

In consistent with the NMR results, the results from the FTIR investigation also showed similar chemical composition of SOM under

different mowing managements. Both suggested that carbohydrates and aromatics were two major constituents of SOM in the surface soils of this semiarid grassland. However, FTIR provided less detailed information of the C functional groups because the intensity of peaks at 2,928 was very weak or nearly absent in all treatments so no obvious treatment effects were observed.

### 4.3 Relationship among net N mineralization, chemical compositions of SOM and microbes

Close correlations were found between net N mineralization and C functional groups. This may be because N-containing compounds, such as

amino acids, amino sugar, pyrimidines, purines or porphyrin structures constituted ca. 2 – 15% of plant mass and some derived from microbial cellular residues (Mathers et al., 2007), and plant mass and microbial cellular residues were also the main sources of SOC in grassland.



Significant correlations between the four groups in the CPMAS [13]CNMR spectra and MBC (Table 8 and 9) and between net N mineralization and MBC (Li et al., 2016) were also observed which confirms that microorganism is the driving force of soil nitrogen transformation. Therefore, soil net N mineralization, soil functional C composition and MBC were related to each other in this study. Our results further showed that the

correlation between C functional groups and MBC was consistent with that between C functional groups and net N mineralization. Previous studies also indicated that chemical composition of SOM significantly influenced soil N mineralization and it was concluded that soils relatively rich in N should also be relatively rich in alkyl C (Stevenson et al., 2016).

Changes of chemical composition of SOC might have been correlated to microbial community across different mowing managements. In grassland or forest ecosystems, recalcitrant C (alkyl C and aromatic C) account for a large proportion of the SOC and it was reported that fungi

played the key role in the decomposition of SON (Boeni et al., 2014; Li et al., 2013). In our study, recalcitrant C (alkyl C and aromatic C) accounted for 40.9 – 47.1% of all functional C, which indicated that fungi might also be critically important for the degradation of these recalcitrant C. Moderate mowing managements increased the abundance and diversity of degrader fungi and mycorrhizal fungi while higher mowing frequency decreased them (Li et al., 2016).by decreasing the microbial biomass (Table 4) and the abundance and diversity of fungal community (Li et al., 2016). This was because fungi could better adapt to the harsh environments (drought and low availability of nutrient) than

bacteria (Andresen et al., 2014; Mariotte et al., 2015). In low fertility grassland soils, a mechanism by which plant productivity could be sustained was through association with mycorrhizal fungi (Northup et al., 1995, 1998). Moderate mowing increased SOC content including labile and recalcitrant C and probably further increased microbial community diversity. It was reported that soil microbial diversity also altered the chemical structure of SOM by degradation and formation of SOM (Baumann et al., 2013) and the diversity of microbial cell membranes was associated with the mobile – $(CH_2)_n$ – structures (Zhang et al., 2015). Thus, the relationship between chemical composition of SOM and

microbial community using modern spectrum technology (CPMAS [13]C-NMR or CPMAS [15]N-NMR) and molecular biological method in this study give us a better understanding of the stability of soil C and N pools.

## 5  Conclusions

Long-term moderate mowing managements (M1/2 and M1) significantly enhanced the accumulation of SOM while the higher frequency mowing practice (M2) limited the accumulation of SOM. Mowing had significant impacts on the fractions and chemical structure of SOM. On

the one hand, moderate mowing (M1/2 and M1) increased both labile and recalcitrant fractions of SOM, improved the stability of SOM by increasing alkyl C, aromatic C functional groups, improving the degree of humification and HB/HI value, while higher frequency mowing practice (M2) had a negative impact on the stability of SOM. These were closely related to soil microbial biomass and SOM mineralization. Moderate mowing managements were beneficial for the degradation of SOM to provide N for plants due to increased MBC and A/OA value and decreased CC/MC, while higher mowing frequency mowing practice had opposite effects, because M2 significantly reduced microbial biomass

and limited SOM humification. Solid CPMAS [13]C-NMR is a powerful technique assessing the complex SOM and it showed that alkyl C and *O*-alkyl C were the dominant chemical composition of SOC under different mowing treatments, followed by aromatic C and carbonyl C. In general, mowing twice a year (M2) is not an appropriate practice according to the stability and balance of SOM in the semiarid grassland while M1/2 and M1 are recommended practices. To better understand the biological mechanisms of SOM composition shifts resulting from different mowing managements, it is necessary to further investigate the microbial community diversity and the relationship between the C functional

groups and microbial community diversity.

*Acknowledgements.* This work was financially supported by the National Key Basic Research Program of China (2014CB138801) and the National Natural Science Foundation of China (41271272 and 41401266).



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



**Figure captions:**

**Fig. 1.** FTIR spectra of bulk SOM under different long-term mowing managements.

**Fig. 2.** CPMAS [13]C-NMR spectra of 2% HF pretreated SOM. **A**, CPMAS [13]C-NMR spectra of 2% HF pretreated SOM under different long-term mowing managements. **B**, detailed C functional groups in different chemical shifts.





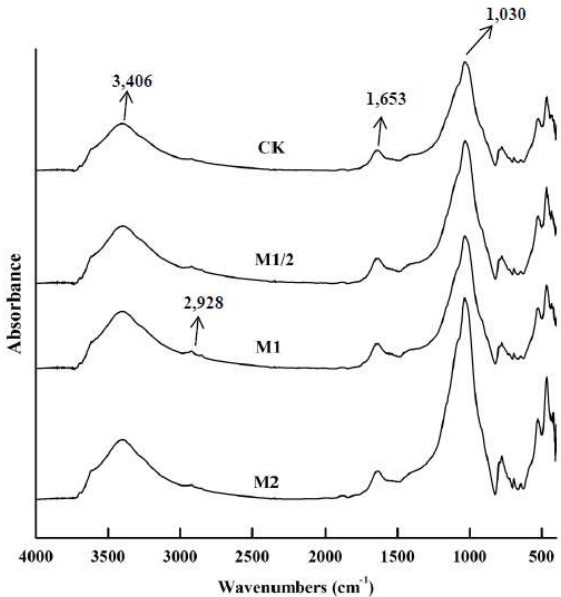

**Fig. 1.** FTIR spectra of bulk SOM under different long-term mowing managements.



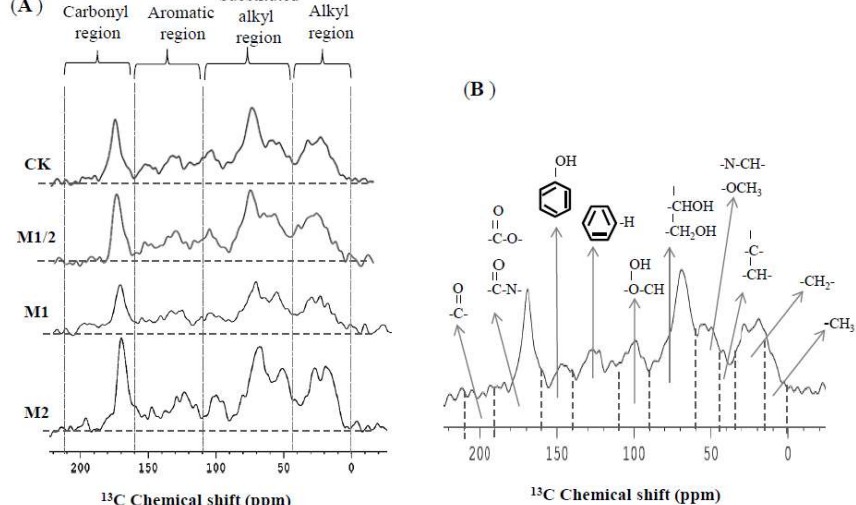

Fig. 2. CPMAS $^{13}$C-NMR spectra of 2% HF pretreated SOM. **A**, CPMAS $^{13}$C-NMR spectra of 2% HF pretreated SOM under different long-term mowing managements. **B**, detail C functional groups in different chemical shifts.





**The list of Tables**





**Table 1**

The description of mowing treatments

| Treatment | Description |
| --- | --- |
| M1/2 | Herbage was harvested once every second year on the 16th, Aug. |
| M1 | Herbage was harvested once every year on the 16th Aug. |
| M2 | Herbage was harvested twice every year on the 23rd, Jun. and on the 12th, Sep. |





**Table2**

Calculation formulas of different [13]C-NMR indexes

| Index | Formula | Reference |
|---|---|---|
| A/OA | alkyl C (45 – 0 ppm) / *O*-alkyl C (110 – 60 ppm) | Maters et al., (2007) |
| CC/MC | carbonhydrate C (90 – 60 ppm) / methoxyl C (60 – 45 ppm) | Zhao et al., (2012) |
| HB/HI | hydrophobic C (45 – 0 ppm + 160 – 110 ppm) / hydrophilic C (110 – 60 ppm + 210 – 160 ppm) | Spaccini et al., (2002) |
| Aliphaticity, % | (alkyl C + Substituted C)*100 / ( alkyl C + substituted C + aromatic C) | |
| Aromaticity, % | aromatic C *100 / ( alkyl C + substituted C + aromatic C) | Zhao et al., (2012) |
| Al/Ar | aliphaticity / aromaticity | |
| Lignin C | phenolic C *4.5 + methoxyl C | |
| Polysaccharide C | 1.2*(*O*-alkyl C – phenolic C *1.5) | Preston et al., (1998) |
| L/P | lignin C/ polysaccharide C | |





**Table 3**

Basic description of soil properties under different mowing treatments

| Treatment | pH | SOC | TN | Olsen P | AN | Net N mineralization |
|---|---|---|---|---|---|---|
| | | g kg$^{-1}$ | | mg kg$^{-1}$ | | mg N g$^{-1}$ |
| CK | 7.3±0.1 a | 17.9±0.6 b | 1.5±0.0 ab | 1.0±0.15 a | 75±0.59 a | 194±3.76 b |
| M1/2 | 7.3±0.0 a | 20.2±1.6 a | 1.5±0.4 ab | 1.3±0.06 a | 86±1.42 a | 176±7.51 b |
| M1 | 7.3±0.1 a | 21.7±0.3 a | 1.7±0.0 a | 1.2±0.03 a | 86±0.00 a | 225±2.51 a |
| M2 | 7.2±0.0 a | 17.8±0.8 b | 1.3±0.0 b | 1.2±0.03 a | 57±0.00 b | 127±7.50 c |

The treatments are introduced in detail in the field experimental designs. The values are the mean ± S.E., n = 3. Abbreviations are as follows:

SOC, soil organic carbon; TN, total nitrogen; Olsen P, Olsen phosphorus; AN, alkali-hydrolysable nitrogen; Net N mineralization, net nitrogen

mineralization. Different lowercase letters in the same column indicate the difference between treatments reaches 5% significant level.





**Table 4**

Effect of different mowing managements on bulk SOM fractions

| Treatment | MBC | WSOC | ROC | MHA | CaHA | HA/SOM |
|---|---|---|---|---|---|---|
| | mg kg$^{-1}$ | | | g kg$^{-1}$ | | % |
| CK | 139.0 ±9.81 b | 98.6 ±9.42 a | 7.3±0.65 a | 6.0±0.76 a | 14.0 ±0.87b | 64.8 a |
| M1/2 | 167.9 ±3.70 a | 42.4 ±3.51 b | 3.1±0.17 bc | 4.6 ±0.76a | 20.5±0.53 a | 72.2 a |
| M1 | 144.6 ±8.09 b | 45.6 ±2.37 b | 3.5±0.20 b | 6.0 ±0.55a | 21.5±0.46 a | 73.0a |
| M2 | 101.3 ±6.23 c | 38.8 ±5.51 b | 2.3±0.12 c | 3.9±0.57 b | 12.9 ±0.89b | 53.1 b |

WSOC, water soluble organic carbon. MBC, microbial biomass carbon. ROC, readly oxidization carbon. MHA, mobile humic acid. CaHA, calcium humic acid. SOM, soil total organic matter. HA=MHA+CaHA.



**Table 5**

Elemental composition of SOM from surface soils in grassland soil with different mowing frequencies

| Treatment | Elemental composition, % | | | | Atom ratios | |
|---|---|---|---|---|---|---|
| | C | H | N | O | H/C | O/C |
| CK | 3.94 b | 0.63 a | 0.38 b | 0.35 a | 0.16 a | 0.09 a |
| M1/2 | 3.95 b | 0.51 b | 0.39 b | 0.28 ab | 0.13 b | 0.07 b |
| M1 | 4.32 a | 0.56 b | 0.41 a | 0.26 ab | 0.13 b | 0.06 b |
| M2 | 3.28 c | 0.49 b | 0.27 c | 0.25 b | 0.15 a | 0.08 a |

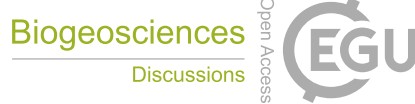

**Table 6**

Percentages of total special spectral area of different functional groups obtained by quantitative CPMAS $^{13}$C-NMR for soil samples from grassland soil with different mowing frequencies (%)

| Treatment | Alkyl C | Substituted alkyl C | | | Aromatics | | Carbonyls |
|---|---|---|---|---|---|---|---|
| | 45 – 0 ppm | 60 – 45 ppm | 90 – 60 ppm | 110 – 90 ppm | 140 – 110 ppm | 160 – 140 ppm | 210 – 160 ppm |
| | Alkyl | *N*-alkyl/methoxyl | *O*-alkyl | di-*O*-alkyl | Aryl | *O*-aryl | Carboxyl and carbonyl |
| CK | 24.6 | 11.2 | 27.0 | 9.4 | 11.7 | 4.6 | 11.1 |
| M1/2 | 27.6 | 12.9 | 22.5 | 8.6 | 13.4 | 5.7 | 9.3 |
| M1 | 27.9 | 13.4 | 22.3 | 8.1 | 13.6 | 5.6 | 9.1 |
| M2 | 25.4 | 12.3 | 21.7 | 7.5 | 12.8 | 6.6 | 13.7 |





**Table 7**

CPMAS $^{13}$C-NMR indices of SOM from surface soils in grassland soils with different mowing frequencies

| Treatment | Lingin-C | Polysaccharide-C | L/P | Aliphaticity | Aromaticity | Al/Ar | A/OA | HB/HI | CC/MC |
|---|---|---|---|---|---|---|---|---|---|
| | % of SOC | | | % | | | | | |
| CK | 36.40 | 13.92 | 2.61 | 79.3 | 20.7 | 3.84 | 0.54 | 0.76 | 2.23 |
| M1/2 | 38.55 | 16.74 | 2.30 | 78.9 | 21.1 | 3.75 | 0.63 | 0.88 | 1.74 |
| M1 | 38.15 | 16.86 | 2.26 | 77.5 | 22.5 | 3.85 | 0.64 | 0.87 | 1.66 |
| M2 | 42.00 | 14.16 | 2.97 | 79.4 | 20.6 | 3.45 | 0.61 | 0.81 | 1.76 |

L/P, lignin/polysaccharide. A/OA, alkyl C/O-alkyl C. HB/HI, hydrophobic C/hydrophilic C. CC/MC = carbonhydrate C/methoxyl C. Al/Ar,

Aliphaticity/Aromaticity.





**Table 8**

Linear correlation coefficients for relationships among different SOM fractions and net N mineralization

|  | SOC | WSOC | MBC | ROC | MHA | CaHA | Net N mineralization |
|---|---|---|---|---|---|---|---|
| SOC | 1 |  |  |  |  |  |  |
| WSOC | 0.11 | 1 |  |  |  |  |  |
| MBC | **0.45** | 0.37 | 1 |  |  |  |  |
| ROC | 0.36 | **0.90** | **0.55** | 1 |  |  |  |
| MHA | **0.48** | 0.43 | 0.45 | **0.92** | 1 |  |  |
| CaHA | **0.89** | 0.41 | 0.34 | **0.81** | 0.23 | 1 |  |
| Net N mineralization | **0.54** | 0.08 | **0.60** | **0.91** | **0.83** | **-0.75** | 1 |

The bold denotes the difference was significant at the level of $P < 0.05$. SOC, soil total organic carbon; The others were the same as Table 4.



**Table 9**

Summary of the linear correlation for relationships between Net N mineralization, MBC and all the C functional groups of SOC determined by CPMAS $^{13}$C-NMR

| Chemical shifts region, ppm | Net N mineralization | | MBC | |
|---|---|---|---|---|
| | r | *P* | r | *P* |
| **Detail assignments** | | | | |
| Alkyl C (45 − 0) | 0.46 | **0.047** | 0.59 | **0.039** |
| *N*-alkyl/methoxyl C (60 − 45) | 0.28 | 0.615 | 0.29 | 0.891 |
| *O*-alkyl C (90 − 60) | 0.37 | 0.429 | 0.27 | 0.992 |
| di-*O*-alkyl C (110 − 90) | 0.47 | 0.326 | 0.59 | **0.027** |
| Aryl C (140 − 110) | -0.24 | 0.798 | 0.73 | **0.011** |
| *O*-aryl C (160 − 140) | -0.94 | **<0.001** | -0.84 | **<0.001** |
| Carbonyl C (210 − 160) | -0.79 | **0.005** | -0.96 | **0.003** |
| **Integrated regions** | | | | |
| Unsubstituted alkyl C (45 − 0) | – | – | – | – |
| Substituted alkyl C (110 − 45) | 0.70 | **0.010** | 0.68 | **0.014** |
| Aromatics (160 − 110) | -0.81 | **0.008** | -0.39 | **0.042** |
| Carbonyls (210 − 160) | – | – | – | – |

n = 20. Substituted alkyl C integrated *N*-alkyl/methoxyl C, *O*-alkyl C and di-*O*-alkyl C. Aromatics integrated aryl C and *O*-aryl C. In the integrated regions, unsubstituted alkyl C and carbonyls were the same as alkyl C and carbonyl C in detailed assignments, respectively.