# Peer review of "Effects of long-term mowing on the fractions and chemical composition of soil organic matter in a semiarid grassland"

_Biogeosciences, 2016_

## Referee Comment (RC1) · Anonymous Referee #1 · 19 Dec 2016

General comments:

Li et al. studied the long term effect of different mowing management practices on SOM properties in semi-arid grassland soil of Inner Mongolia. The authors used FTIR and 13C-NMR spectroscopy to characterize SOM. Further they analyzed certain SOM fractions and bulk soil parameters.

The topic of land management intensity effects on SOM fits well to the scope of the Journal. The used methods are adequate to characterize SOM and their combination should be of particular interest. The results are interesting; however, there are some issues which need improvement before publication of the manuscript.

[Figure]

Specific comments:

1. From the abstract it does not become clear that C and some N parameters are of interest; abbreviations are not mentioned (NMR ratios, MBC) or are introduced without using them (CK).

2. The introduction needs to provide more information on (the used) SOM fractions and other characteristics (e.g. O-alkyl C) and what can be deduced from them. What information do we gain from FTIR and 13C NMR? What is stable what is labile SOM and which one is good for what?

3. The authors should stick to abbreviations of mowing treatments rather than trying to word it. This will a) be more precise (e.g. line 204) and b) will make it easier to read the text.

4. Instead of 'CK, unmown' I suggest the abbreviation M0 for the control.

5. I'm wondering whether storage at 4°C is not too warm for samples taken from a region with an annual mean temperature of 0.5°C (line 57)?

6. Only 1 replicate was analyzed by FTIR and NMR but a) it does not become clear which one it was (start or end of time line?, mixture of replicates?); b) this unfortunately has to result in an interpretation without statements on 'significant difference'.

7. The discussion is nicely separated into interesting headlines. However, the authors do not meet the reader's expectations in the text then. This may be due to a) not clearly separating between discussion about SOM N and SOM C and b) not discussing the mowing practices results in a certain order and c) trying to draw conclusions from tables which cannot be deduced from those E.g. Table 8: It is not clear which data are shown: are these r values for all samples taken together or is this just one mowing frequency, and if so, which one? Shifts due to mowing frequency cannot be seen from this table. The authors would need to create graphs of relationships (e.g. SOM and WSOC) in which points represent mowing treatments. Only from those results conclusions could

be drawn on mowing effects.

8. Instead of intensively discussing the calculated SOM ratios etc. the authors elaborate on N mineralization, microbial community structure and mycorrhizal fungi. These are important factors but they need to be better imbedded into the main topic of interest: SOM. Too much speculation or repetition of results from previous studies should be avoided.

9. It would be helpful, if result values were compared to literature results for similar grassland systems.

10. In my opinion plant species, richness and productivity of the different mowing treatments are very important parameters in the context of this manuscript and should be shown as well. An interesting question then may be answered more easily: How does different vegetation affect SOM characteristics/are the differences in SOM due to differences in vegetation parameters?

11. The English needs some improvement in particular with regard to tenses.

More detailed comments:

Introduction

Line 38: It does not become clear in which context 'particle size' plays a role in this manuscript.

Line 49: Baumann et al. 2013 do not show any element or FTIR analyses; however, Baumann et al. 2016 do (Geoderma 278, 49–57).

Material and Methods

It does not become clear for which analyses air dried soil was used and for which field moist soil was used.

Line 65ff: unclear; when did the experiment start, when did it end?, what are the 5

treatments, what are the 3 treatments?

Line 69: I suggest to present months without exact days since they vary anyway.

Line 76: replace 'transformed' by 'transported'

Line 77: delete 'soil sample'

Line 79: Was the soil dried until constant mass or for 16h?

Line 84: who/what is 'they'?

Line 86: give 'g' instead of 'rpm'

Line 87: give details for the TOC analyzer

Line 89: change to freeze-drying

Line 94: supplementary material and C/N are not visible to me

Line 108: insert 'state 13C' before 'NMR'

Line 111: separate hexamethylbenzene properly, if necessary

Results

Only 'effects' or 'no effects' should be described but not 'little effects' if there is no statistical difference (e.g. line 120).

Line 11: define P<0.05 here and delete if from results and discussion sections.

Line 135: remove citation form here

Line 150 ff: move to discussion

Line 163-167: delete

Line 169: how was the 1 replicate prepared, please state here

Table 6: delete '%' in caption

Line 194: do you mean 'microbial biomass' or 'MBC'?

Discussion

Often the sentences are just lined up without any connection (e.g. line 204ff). Please create a story.

Line 204: what is a 'the overall and significant increased trend of SOM'? unclear; do you mean SOM content? Increased with what? Is it a trend or is it significant?

Line 206: do you mean number of plant species?

Line 211: not deducible from Table 8

Line 228: only refer to M1/2 and M1

Line 233: delete 'charcoal' as it is not relevant here; 'concentration' of aromatics was not analyzed

Line 234: or enhanced by plants with higher aromatic content?

Line 239: Table 3 does not show microbial biomass

Line 248 ff: move ratio explanation to methods section

Line 277: can the previous studies be confirmed by own results?

Line 280: what is SON?

Line 286: what are mycorrhizal fungi doing in the context of this manuscript?

Line 302: M2 shows a higher stability of SOM: would this perhaps be better for storing C in soil?

References

Cardinale et al. 2012 is missing

Chong et al. 2014 is missing or misspelled

---

## Referee Comment (RC2) · Anonymous Referee #2 · 1 Feb 2017

rassland sustains the feed for livestock and possesses the second largest C pool following forest. This study characterized the structure and composition of soil organic matter in grassland soils received long-term mowing at different frequency, using the traditional method combined with advanced spectroscopy (13C-NMR and FTIR) techniques. The results revealed that the medium-frequency mowing could significantly enhance the SOM accumulation and increased the stability of SOM while high-frequency mowing (twice a year) went contrarily. The findings are interesting considering the ecological function of grassland as important C pool and their service function for livestock, and of significance to guide the grassland management. The study is conducted well and the paper is clearly presented, while English in some sentences could be further

polished (like line 13, line 175 etc.), and the significance of the finding could be further highlighted.

Specific comments:

1. The information on treatments detail in Table 1 could be included in Table 3, and Table 1 is not necessary.

2. The measurements for different parameters of SOM in this study were conducted only for one sampling time point. Supplying some annual regular investigation data such as SOC content etc. will be helpful to solidify the conclusion of the study.

---

## Author Comment (AC1) · 3 Mar 2017

Comments from referee 1 and the responces to them from authors

General comments from referee 1:

Li et al. studied the long term effect of different mowing management practices on SOM properties in semi-arid grassland soil of Inner Mongolia. The authors used FTIR and 13C-NMR spectroscopy to characterize SOM. Further they analyzed certain SOM fractions and bulk soil parameters. The topic of land management intensity effects on SOM fits well to the scope of the Journal. The used methods are adequate to characterize SOM and their combination should be of particular interest. The results are interesting; however, there are some issues which need improvement before publication of the manuscript.

Authors: Thanks a lot for your comments and we have tried our best to solve the issues you mentioned above and to improve the MS.

Specific comments:

1. From the abstract it does not become clear that C and some N parameters are of interest; abbreviations are not mentioned (NMR ratios, MBC) or are introduced without using them (CK).

Authors: We have checked and reedited the abstract. Please see the highlight part in the abstract of the MS.

2. The introduction needs to provide more information on (the used) SOM fractions and other characteristics (e.g. O-alkyl C) and what can be deduced from them. What information do we gain from FTIR and 13C NMR? What is stable what is labile SOM and which one is good for what?

Authors: Thanks for your valuable suggestions. We have added this part in the introduction part. Please see P2 – 3, L40, L46 – 48 and L56 – 61.

3. The authors should stick to abbreviations of mowing treatments rather than trying to word it. This will a) be more precise (e.g. line 204) and b) will make it easier to read the text.

Authors: Thanks for your valuable suggestions and We have used the abbreviations of mowing treatments through the MS.

4. Instead of 'CK, unmown' I suggest the abbreviation M0 for the control.

Authors: Thanks for your constructive suggestion and we have changed 'CK' to 'M0' through the MS.

5. I'm wondering whether storage at 4°C is not too warm for samples taken from a region with an annual mean temperature of 0.5°C (line 57)?

Authors: 0.5°C is just the annual mean temperature and generally, 4°C is the temperature that most of microbes have no activity. Actually, the soil samples were treated and the corresponding indexes were measured as soon as we took them back to lab and then they were stored -20°C.

6. Only 1 replicate was analyzed by FTIR and NMR but a) it does not become clear which one it was (start or end of time line? mixture of replicates?); b) this unfortunately has to result in an interpretation without statements on 'significant difference'.

Authors: We initially measured the mixture of replicates as it was reported that data of the replicates were similar in many previous studies and therefore these studies just measured the mixed samples. Now we have measured all of the three replicates and the results also prove that the values of three replicates are similar because the values of standard errors are very small. Please see Table 5.

7. The discussion is nicely separated into interesting headlines. However, the authors do not meet the reader's expectations in the text then. This may be due to a) not clearly separating between discussion about SOM N and SOM C and b) not discussing the mowing practices results in a certain order and c) trying to draw conclusions from tables which cannot be deduced from those E.g. Table 8: It is not clear which data are shown: are these r values for all samples taken together or is this just one mowing frequency? and if so, which one? Shifts due to mowing frequency cannot be seen from this table. The authors would need to create graphs of relationships (e.g. SOM and WSOC) in which points represent mowing treatments. Only from those results conclusions could be drawn on mowing effects.

Authors: Thanks for your valuable comments and we have separated SOM N and SOM C. We did not show the graphs of relationship between different SOM fractions instead of a table (Table 7). On the one hand, we have showed the data of different

SOM fractions contents under different mowing treatments in Table 2 and 3; On the other hand, we measured 7 kind of SOM fractions it was not practical to use graphs to show the relationships between the 7 fractions. In addition, we have added some explanations in the tables to make the expression clearer. Please see the Discussion part and Table 7 and 8.

8. Instead of intensively discussing the calculated SOM ratios etc. the authors elaborate on N mineralization, microbial community structure and mycorrhizal fungi. These are important factors but they need to be better imbedded into the main topic of interest: SOM. Too much speculation or repetition of results from previous studies should be avoided.

Authors: Thanks for your constructive comments. We have deeply discussed these important factors and moderately reduced the speculation or repetition of results to elaborate the level of the discussion part in the MS. Please see the Discussion part.

9. It would be helpful, if result values were compared to literature results for similar grassland systems.

Authors: There has been little literature documented the chemical structures of SOM in grassland ecosystem which are widely studied in forest and farming ecosystems till now. Therefore, following your comments we have compared our results with the results from other ecosystems to make our study more implicative and significant. Please see the Discussion.

10. In my opinion plant species, richness and productivity of the different mowing treatments are very important parameters in the context of this manuscript and should be shown as well. An interesting question then may be answered more easily: How does different vegetation affect SOM characteristics/are the differences in SOM due to differences in vegetation parameters?

Authors: Many studies conducted in ecology have reported plant species, richness

and productivity are closely related with the SOM content. Our study was based on a long-term field trial and we did not investigate plant species, richness and productivity of the different mowing treatments as the results have been reported in a recent study in this long-term field trial (Baoyin et al., 2014). We have cited this paper in our study to explain some results in our study. However, the goal of our study is to quantificationally evaluate the stability of SOM from the view of chemical structures by advanced techniques (CPMAS 13C-NMR) under different mowing managements, and we have reedited the MS to better show our study goals.

11. The English needs some improvement in particular with regard to tenses.

Authors: We have asked the English native speaker (Prof. Di) to seriously edit the MS to improve the English and the tenses were corrected.

More detailed comments from referee 1 and responces from authors:

Introduction

Line 38: It does not become clear in which context 'particle size' plays a role in this manuscript.

Authors: We have reedited the contents. Please see P2, L44-45.

Line 49: Baumann et al. 2013 do not show any element or FTIR analyses; however, Baumann et al. 2016 do (Geoderma 278, 49–57).

Authors: We feel so sorry that we made a mistake on this reference. We have checked the MS and added this reference (Baumann et al. 2016) in the MS.

Material and Methods

It does not become clear for which analyses air dried soil was used and for which field moist soil was used.

Authors: We have rewritten the related sentences and made the question clear.

Line 65ff: unclear; when did the experiment start, when did it end? what are the 5 treatments, what are the 3 treatments?

Authors: We made some mistakes on the statements and we have revised them. The long-term experience has been started since 2001 and has been conducted till now.

Line 69: I suggest to present months without exact days since they vary anyway.

Authors: Changed and please see P3, L86 – 88.

Line 76: replace 'transformed' by 'transported'

Authors: Changed.

Line 77: delete 'soil sample'

Authors: Deleted.

Line 79: Was the soil dried until constant mass or for 16h?

Authors: Yes.

Line 84: who/what is 'they'?

Authors: They indicate water soluble organic carbon and readily oxidized carbon contents and we have changed "they" into "WSOC and ROC contents".

Line 86: give 'g' instead of 'rpm'

Authors: Changed.

Line 87: give details for the TOC analyzer

Authors: Changed.

Line 89: change to freeze-drying

Authors: Changed.

Line 94: supplementary material and C/N are not visible to me

Authors: We feel so sorry for the missing of Table S2, which showed the C/N ratios.

Line 108: insert 'state 13C' before 'NMR' Authors: Changed.

Line 111: separate hexamethylbenzene properly, if necessary

Authors: The 13C-NMR results have separated hexamethylbenzene properly in our study.

Results

Only 'effects' or 'no effects' should be described but not 'little effects' if there is no statistical difference (e.g. line 120).

Authors: Changed.

Line 11: define P <0.05 here and delete if from results and discussion sections.

Authors: Changed.

Line 135: remove citation form here

Authors: Removed.

Line 150 ff: move to discussion

Authors: Changed.

Line 163-167: delete

Authors: Deleted.

Line 169: how was the 1 replicate prepared, please state here

Authors: We have measured the tree replicates. The means of tree replicates and statistical analysis were showed in Table 5 and 6.

Table 6: delete '%' in caption

Authors: Changed.

Line 194: do you mean 'microbial biomass' or 'MBC'?

Authors: MBC is the abbreviation of microbial biomass carbon. The increase of MBC indicates the increase of microbial biomass.

Discussion

Often the sentences are just lined up without any connection (e.g. line 204ff). Please create a story.

Authors: Thanks, and we have tried our best to the discussion more logical to create a story.

Line 204: what is a 'the overall and significant increased trend of SOM'? unclear; do you mean SOM content? Increased with what? Is it a trend or is it significant?

Authors: Thanks a lot for your comments. We have rewritten the discussion part to make them clear. Please see P7, L220 – 221.

Line 206: do you mean number of plant species?

Authors: Yes, and we have made it clear in the MS.

Line 211: not deducible from Table 8

Authors: We have corrected the sentences to make the meaning understandable.

Line 228: only refer to M1/2 and M1

Authors: Yes, our study showed compared with CK (unmown), M1/2 and M1 were the moderate mowing practices while M2 was not suitable for a long-term mowing management as it resulted the significant reductions of SOM content, MBC content and net N mineralization rate.

Line 233: delete 'charcoal' as it is not relevant here; 'concentration' of aromatics was not analyzed

Authors: Changed.

Line 234: or enhanced by plants with higher aromatic content?

Authors: Generally, what you suggested was possible. However, our results just showed that humic acid MBC increased in the moderate mowing treatments, so we deduced that the humifaction process was enhanced.

Line 239: Table 3 does not show microbial biomass

Authors: We make a mistake and it should be Table 4 originally.

Line 248 ff: move ratio explanation to methods section

Authors: We have followed your suggestions.

Line 277: can the previous studies be confirmed by own results?

Authors: Yes, our results also showed that soils relatively rich in N was also rich in alkyl C. The key reason was alkyl C was the degradation production of SOM, available N increased while SOM was degraded as the N mineralization increased, which would be benefit for the increase of plant productivity and soil N input by more leguminous plant N fixation in turn.

Line 280: what is SON?

Authors: 'SON' means "soil organic nitrogen", and we have reedited it in the MS.

Line 286: what are mycorrhizal fungi doing in the context of this manuscript?

Authors: In our recent study conducted in this long-term filed trial showed that moderate mowing increased the abundance of soil mycorrhizal fungi. In the low fertility soils, mycorrhizal fungi could help plants absorb the moisture and nutrients which were far from the rhizospheres and thus sustain or increase the plant productivity and thus sustain or increase soil nutrients by plant residues and roots biomass and root exudates
.

Line 302: M2 shows a higher stability of SOM: would this perhaps be better for storing C in soil?

Authors: The results showed that M2 was not benefit for the accumulation of SOM and it showed a reduction of SOM content. As well, the microbial community reduction and available nutrients lack in M2 resulted in the low plant productivity, which resulted a vicious cycle of plant-soil-microbial and further destroyed the grassland ecosystem balance. Our samples were collected in the Oct. when the temperature was low and many microbes had low activity. Once the temperature rises, the relatively rich easily degradable chemical structure of SOC will be degraded and the SOC showed a negative budget. Therefore, our study showed that M2 not only decreased the SOM content, but also was potentially adverse to the stability of SOM.

References

Cardinale et al. 2012 is missing

Authors: We feel very sorry for this mistake and we have added this reference in the reference list.

Chong et al. 2014 is missing or misspelled

Authors: We feel very sorry for this mistake and we have corrected it.

---

## Author Comment (AC2) · 3 Mar 2017

General comments referee 1 and the responces to them from authors:

Grassland sustains the feed for livestock and possesses the second largest C pool following forest. This study characterized the structure and composition of soil organic matter in grassland soils received long-term mowing at different frequency, using the traditional method combined with advanced spectroscopy (13C-NMR and FTIR) techniques. The results revealed that the medium-frequency mowing could significantly enhance the SOM accumulation and increased the stability of SOM while high-frequency mowing (twice a year) went contrarily. The findings are interesting considering the ecological function of grassland as important C pool and their service function for livestock,

and of significance to guide the grassland management. The study is conducted well and the paper is clearly presented, while English in some sentences could be further polished (like line 13, line 175 etc.), and the significance of the finding could be further highlighted.

Authors: Thanks a lot for your valuable comments. We have asked the English native speaker (Prof. Di) to seriously edit the MS thoroughly to improve the English language. The introduction and discussion were majorly revised to further highlight the significance of the findings, please see these two parts of MS.

Specific comments referee 2 and the responces to them from authors:

1. The information on treatments detail in Table 1 could be included in Table 3, and Table 1 is not necessary

Authors: We have deleted the Table 1 and the corresponding contents is removed to the experimental design part of the MS. Please see P3, L86 – 88.

2. The measurements for different parameters of SOM in this study were conducted only for one sampling time point. Supplying some annual regular investigation data such as SOC content etc. will be helpful to solidify the conclusion of the study.

Authors: Although the field trial has been started since 2001, the program supported our study began in Sep., 2013 when we conducted this study based on the long-term trial. The sampling time was decided properly to evaluate the impacts of different mowing practices on the stability of SOM by analyzing the differences between the mowing treatments and CK (unmown). To avoid the influences of grass, we chose to collect soil samples in Oct. Therefore, we did not show the investigation data at other time point in this study. However, we measured the MBC and SOC contents and the chemical composition of SOM in the soil samples collected in Oct., 2014, which showed the similar results. Given that we had not measured the contents of other SOM fractions (WSOC, ROC, MHA and CaHA), we did not show the data in this study.